# Towards Clinical Implementation of Adeno-Associated Virus (AAV) Vectors for Cancer Gene Therapy: Current Status and Future Perspectives

**DOI:** 10.3390/cancers12071889

**Published:** 2020-07-14

**Authors:** Ulrich T. Hacker, Martin Bentler, Dorota Kaniowska, Michael Morgan, Hildegard Büning

**Affiliations:** 1Department of Oncology, Gastroenterology, Hepatology, Pulmonology, and Infectious Diseases, University Cancer Center Leipzig (UCCL), Leipzig University Medical Center, 04103 Leipzig, Germany; dorota.kaniowska@izi-extern.fraunhofer.de; 2Institute of Experimental Hematology, Hannover Medical School, 30625 Hannover, Germany; Bentler.martin@mh-hannover.de (M.B.); Morgan.michael@mh-hannover.de (M.M.); 3REBIRTH Research Center for Translational Regenerative Medicine, Hannover Medical School, 30625 Hannover, Germany; 4German Center for Infection Research (DZIF), Partner Site Hannover-Braunschweig, Inhoffenstraße 7, 38124 Braunschweig, Germany

**Keywords:** adeno-associated virus (AAV) vectors, transductional targeting, transcriptional targeting, hallmarks of cancer, apoptosis, tumor angiogenesis, immunotherapy, vaccine, CAR-T cells

## Abstract

Adeno-associated virus (AAV) vectors have gained tremendous attention as in vivo delivery systems in gene therapy for inherited monogenetic diseases. First market approvals, excellent safety data, availability of large-scale production protocols, and the possibility to tailor the vector towards optimized and cell-type specific gene transfer offers to move from (ultra) rare to common diseases. Cancer, a major health burden for which novel therapeutic options are urgently needed, represents such a target. We here provide an up-to-date overview of the strategies which are currently developed for the use of AAV vectors in cancer gene therapy and discuss the perspectives for the future translation of these pre-clinical approaches into the clinic.

## 1. Introduction

Cancer represents a major health problem worldwide. In 2018, the global incidence was as high as 17,000,000 new cases and more than 9,400,000 patients died from cancer (https://gco.iarc.fr; accessed 16 March 2020). Accordingly, cancer is the second highest cause of death worldwide. In their seminal review in 2000, Hanahan and Weinberg [1] described six hallmarks of cancer. With an increasing understanding of cancer biology, these initial six hallmarks were updated to include four additional ones [2]. Importantly, these cancer hallmarks served as a blueprint for the development of new treatment strategies, an impressive number of which has made their way from preclinical development to the clinic (Table 1).

Cancer research in the last 50 years has revealed that cancer represents a disease that is characterized by dynamic changes in the genome. Consequently, gene therapy has long been proposed as a promising technology for the development of cancer treatment strategies. More recently, as the role of the tumor microenvironment has been fully recognized as a central player in tumor progression and metastasis, cells within the tumor microenvironment (i.e., vascular cells, immune cells) emerged as novel targets. Finally, the latest developments in cancer immunotherapy have opened new exciting avenues for the development of cancer gene therapy. In addition to their ability to introduce novel genes into target cells, a number of gene therapy vectors are prime candidates for the development of anti-cancer vaccines or oncolytic virotherapy, and some have already reached the clinic [3].

Here we will focus on the role of adeno-associated virus (AAV) vectors, by giving a comprehensive overview over latest developments and the perspectives we envision for the use of this vector system in the context of the rapidly evolving field of cancer research and treatment.

## 2. Adeno-Associated Virus and AAV Vector Technology

### 2.1. From Virus to Vectors—A Brief Introduction into the AAV Vector System

Treating a genetic disease by counteracting the malfunctioning gene through adding a functional copy (gene addition therapy) or by exchanging the defective gene (replacement therapy) represents the basic concept of gene therapy. In cancer with its unique disease background and involvement of non-malignant cells, additional strategies are developed. Examples for the latter are the overexpression of anti-angiogenic factors (anti-angiogenesis) or genetic engineering of T lymphocytes or NK cells to express chimeric antigen receptors (CARs). Delivery systems to transport the “therapeutic agent” to the target cell are required for all of these approaches. AAV vectors represent a delivery system that is applied with great success in basic and pre-clinical research as well as in clinical applications, particularly in vivo. Currently more than 110 human clinical trials are listed at ClinicalTrials.gov (https://clinicaltrials.gov; accessed 16 March 2020) that use AAV vectors in the context of monogenetic inherited diseases. Reports on therapeutic benefit for patients and the excellent safety record of AAV vectors resulted in three marketing approvals in Western countries (Glybera, Luxturna, Zolgensma) [4].

AAV vectors are derived from adeno-associated viruses. AAV are replication-deficient members of the *Parvoviridae* forming the genus *Dependoparvovirus* [5]. They were discovered in the 1960s as a laboratory contamination of adenovirus (AdV) preparations [5]. This close association, which is also found in nature, is based on the dependency of AAV on helper virus function for progeny production [6]. Later, endogenous AAV sequences were also detected in various vertebrates, including humans and non-human primates [7,8,9].

Although naturally occurring AAVs differ in the amino acid sequence of their capsid proteins and in the length of their genomes, all AAVs are composed of an icosahedral protein capsid assembled from 60 subunits that protects the single-stranded DNA genome and mediates cell infection [10,11]. Antigenic drift forced evolution of serotypes which particularly differ at those sites involved in cell attachment and entry [12]. This portfolio of serotypes forms a solid basis for vector development. So far, no disease has been directly linked to AAV infection. Additional advantageous features for gene therapy are high stability, low immunogenicity, relative ease of vector construction, the ability to transduce dividing as well as non-dividing cells and the broad range of cell types that are transduced upon exposure. Compared with other viral vector systems such as adenoviral or retro-/lentiviral vectors, however, the coding capacity of AAV vectors with approximately 5 kb is low. In addition, AAV vectors lack an integrase activity (in contrast to retro-/lentiviral vectors) and are therefore considered as non-integrating. Nevertheless, strategies to integrate AAV vector genomes site specifically [13,14] or to maintain episomes in proliferating cells by incorporation of autonomous replicating units into the AAV vector genomes [15] have been developed.

In first generation AAV vectors, serotype capsids are loaded (packaged) with vector genomes containing a transgene expression cassette instead of the viral genes. The sole wildtype (wt) sequence that remains in cis in AAV vector genomes are the AAV-specific inverted terminal repeats (ITRs). ITRs form the 5′ and 3′ ends of both viral and vector genomes and serve as origin of replication and packaging signals. The AAV-specific genes, *rep* and *cap*, which encode for a family of multifunctional non-structural proteins (Rep proteins), the three capsid proteins (viral protein (VP) VP1, VP2 and VP3) and assembly activating proteins (AAP) are provided in addition to the helper virus function in trans during vector production (for details on AAV production see [16]). Pseudo-packaging is possible for most of the serotypes, i.e., vector genomes flanked by ITRs of AAV2 are packaged into non-AAV2 capsid, which enables an easy and fast switch between serotypes and thus allows use of the full potential of serotype-specific features [17].

### 2.2. AAV Vector-Mediated Transduction

Transduction is initiated by attachment of AAV vector particles to glycans expressed on the cell surface, such as heparan sulfate (AAV2, 3, 6, 13), galactose (AAV9) or sialic acid (AAV1, 4, 5, 6) [18]. This first contact primes the particle to bind to serotype-specific internalization receptors followed by clathrin-mediated endocytosis. Vectors are transported within the endosomal system towards the cell nucleus [5]. Upon endosomal acidification, capsids become targets of endosomal enzymes and undergo a conformational change that exposes a phospholipase A2 (PLA2) domain as well as nuclear localization sequences (NLS), which are part of the N′-termini of VP1 (PLA2 plus NLS) and VP2 (NLS) [19]. PLA2 activity leads to release of AAV particles from the endosomal compartment, while NLS might foster nuclear uptake. Neither the mechanistic details nor the exact site of particle uncoating are currently known. Nevertheless, the conversion of single-stranded vector genomes into DNA double-strands occurs in the cell nucleus, providing the template for transcription and thus transgene expression. Additional cell entry routes were described that are maybe used in a cell type- and/or serotype-dependent manner, such as caveolae-mediated endocytosis or the CLIC/GEEC pathway [20]. Since receptor choice and mode of entry impacts efficiency of uptake and intracellular processing, some particles might be shuttled into “non-infectious” pathways and are—despite successful uptake—unable to contribute to cell transduction. On the other hand, modifying receptor usage by capsid engineering will lead to vectors tailored for efficient transduction and—depending on the receptor that has been chosen—for cell-type selective transduction.

### 2.3. In Vivo Gene Therapy with First Generation AAV Vectors—Blueprints for Cancer Gene Therapy

The liver and central nervous system (CNS) are main targets in human clinical trials for patients with monogenetic diseases [4]. Based on their natural tissue preference (tropism) AAV2, 5, 8, 9 and 3B are used in human liver-directed gene therapies, while AAV2, 9, rh10 and rh8 are applied either intravenously or locally in CNS-related therapies. Since AAV vectors developed for liver and CNS gene therapy are currently also pre-clinically used in the context of tumors arising from liver [21,22] or CNS [23,24], the basic developments and distinct features of these targets will be described in more detail in the following sections.

#### 2.3.1. Liver-Directed Gene Therapy

More than 400 rare monogenic disorders associated with the liver are described [25]. In addition, the liver can be affected by tumor formation either originating from liver-resident cells such as hepatocytes or from tumor cells derived from distant tumors that engraft in the liver to form metastasis. From a therapeutic perspective, the liver is unique due to its important role in detoxification, metabolism and immunology [25]. The liver is also unique as it contains 10–15% of the whole blood volume of an individual [25]. For in vivo transduction of hepatocytes, AAV vectors are delivered via the hepatic artery [26] or even the peripheral vein [27]. Liver-directed gene therapy initially focused on Hemophilia B, a blood clotting disorder. The Nathwani trial (NCT 00979238) initiated in 2010 reported long-term clinical benefit with restoration of up to 6% of normal FIX levels, as well as a reduction of more than 90% of bleeding events and use of prophylactic FIX [28]. Constant improvements in study and vector design enables even reaching stabilized FIX plasma levels of up to 20–44% (reviewed in [29]). Similar promising results were reported for Hemophilia A (reviewed in [29]).

#### 2.3.2. CNS-Directed Gene Therapy

The most recently approved AAV vector-based gene therapy in the US and some European Union Member states is Zolgensma. It is applied intravenously and uses AAV9 vectors to deliver the survival motoneuron 1 (SMN1) gene to motor neurons in the CNS for patients with SMA type 1 [4]. Mutations in SMA1 result in degeneration of alpha motor neurons leading to progressive muscle weakness and wasting [30,31]. AAV9 was chosen due to the ability to cross the Blood–Brain-Barrier (BBB) when applied intravenously followed by motor neuron transduction [30,31]. Although the level of transduction is considerably low compared with tissues outside the CNS, particularly liver, a single intravenous infusion was sufficient to extend survival and improve motor function in SMA patients [31]. Interestingly, AAVrh10 was also shown to cross the BBB in mice and non-human primates [32,33], and thus can be considered as alternative to AAV9. Importantly, this unique feature of AAV9, AAVrh10 and maybe a few additional serotypes is of interest for cancer gene therapy as well, since it would allow the targeting of brain tumors via a minimally invasive route. However, additional measures as discussed below need to be applied to restrict transduction to the malignant cell type. Alternative routes of application in CNS-directed gene therapy, including approaches developed in the context of cancer, are intraparenchymal, intrathecal and intracerebroventricular [30]. Vector doses are substantially lower, and the choice of serotypes can be expanded beyond AAV9 and AAVrh10. Cell-types that are transduced in the CNS include neurons, glia cells, astrocytes and oligodendrocytes [30].

### 2.4. Optimizing the AAV Vector System for Cancer Gene Therapy

With regard to cancer therapy, ideally gene transfer as well as expression should be restricted to the cell-type of interest such as the malignant cell or distinct cells of the tumor microenvironment like macrophages, dendritic cells (DC), endothelial cells or fibroblasts. In addition, it would be advantageous to optimize efficacy to minimize the vector dose that needs to be applied and to be equipped for a possible re-application scenario. To reach these goals, both the capsid and the genome of AAV vectors have become targets for engineering. In particular, transcriptional and post-transcriptional targeting as well as transductional targeting strategies are developed (Figure 1).

#### 2.4.1. Focus on the AAV Vector Genome—Achieving Transgene Expression in Cells of the Tumor Microenvironment

The natural AAV genome conformation is a single-stranded DNA that needs to be converted to a DNA double strand prior to initiation of transcription. Conversion occurs either by interstrand hybridization (AAV vectors deliver genomes of sense and antisense conformation with equal efficiency) or by second-strand synthesis via cellular polymerases. While conversion occurs quite efficiently in malignant cells [34] presumably because members of the DNA damage response pathway do not interfere with the second-strand synthesis step [35], self-complementary (sc) AAV vectors were developed to enable or enhance expression in non-malignant cells [36]. These vectors contain a genome capable of intra-strand hybridization and thus form a DNA double strand upon being released from the viral capsid (for details on scAAV genomes see [37]). Consequently, a faster onset of transgene expression especially in vivo was observed and cells incapable of second-strand synthesis became susceptible towards AAV vector-mediated transduction. The advantage of scAAV vectors compared to conventional vectors for turning dendritic cells (DC) into smart vehicles in cancer immunotherapy becomes obvious when reviewing results obtained for AAV 1 to 6 vectors on bone-marrow derived DC [38] or for AAV1 and 2 on plasmacytoid DC, myeloid conventional DC and Langerhans cells [39].

#### 2.4.2. Focus on the AAV Vector Genome—Achieving Specificity by Transcriptional and Post-Transcriptional Targeting

A broadly applied strategy to restrict expression of a therapeutic transgene is transcriptional targeting. Prime examples are the use of tumor-specific promoters such as carcinoembryonic antigen (CEA), telomerase reverse transcriptase (TERT) or Survivin promoters, which are active in cancer cells, but silent in differentiated healthy cells [40,41]. For the TERT promoter, also variants with activity enhancing mutations that developed in malignant cells were identified [42]. Furthermore, tissue-specific promoters are in use, which are active in both the malignant cells and the healthy counterparts, but not in other tissues [40]. Since cells react to environmental stimuli by activating or repressing transcription, stress factors present in solid tumor masses such as hypoxia, lack of glucose and low pH, which are all hallmarks of the tumor microenvironment, have been successfully hijacked to restrict transgene expression in cancer therapy. Specifically, in contrast to healthy tissue, 90% of the solid cancer mass is in a hypoxic condition, which enables the use of hypoxia responsive elements within enhancers or promoters [43]. Hypoxia, but also other tumor-specific stress factors induce the human HSP70B promoter which therefore appears suited to control transgene expression from vector genomes in malignant cells as well as cells from the tumor microenvironment [44]. Similarly, the grp94 promoter, which is activated in response to glucose starvation in physiological conditions, was shown to be strongly activated in a variety of tumors and—importantly—in macrophages of the tumor boarder zone [45]. Interestingly, a preferred expression in the tumor compared to healthy tissue was observed for the constitutively active cytomegalovirus promoter (CMV) when using low vector doses: while enhanced vector uptake in the tumor due to leaky vessels likely contributes to this finding, an at the first glance unexpected effect, the enhanced activity of Nuclear Factor kappa B (NF-κB) in tumors, which binds and activates the CMV promoter, is proposed to play a major role [43]. In addition to the rich body of work that has been done to identify and vectorize natural promoters for tissue and cell-type selective expression, synthetic promoters are also developed. The latter are novel combinations of natural transcription factor binding sites, which in the new architecture are capable of responding to the microenvironmental needs of a given cell type or tissue in an optimal fashion. Seminal work of the Vandendriessche and Chuah lab demonstrated the potency of this strategy in the context of AAV vectors. Specifically, by combining muscle-specific transcriptional cis regulatory modules (CRMs) with a muscle-specific promoter, a 400-fold increase in muscle-specific expression was obtained while off-target expression remained at background levels [46]. Usage of liver-specific promoters optimized in this way resulted in supraphysiologic FIX activity (400%) [47]. A further, very powerful strategy to convey cell-type specificity targets the mRNA. This post-transcriptional targeting strategy relies on microRNA, non-coding RNAs, which mediate mRNA degradation upon binding to the corresponding target sequence [48]. Some members of the microRNA family are expressed in a tissue-, cell lineage-, differentiation- or disease-specific pattern and thus appear as perfect tools to degrade transgene-specific mRNA expressed in healthy, but not in malignant cells. For this, vector genomes are equipped—frequently with multiple copies—of microRNA targeting sites placed in the 3′ untranslated region of the transgene expression cassette. A prominent example is miRNA122a. This miRNA is exclusively expressed in hepatocytes, but not in other tissues, and is downregulated in hepatocellular carcinoma (HCC), i.e., upon malignant transformation of hepatocytes [48]. This might be due to the hyperactivity of Myc, which—as reported by Chang and colleagues [49]—causes widespread repression of miRNAs. The incorporation of miR-122a targeting sites in AAV vector genomes prevented transgene expression from hepatocytes, while transgene production in hepatocellular carcinoma was not affected [50]. Since liver is the main off-target for AAV and other viral vectors, miR-122a targeting sites are frequently explored for liver de-targeting in the context of cancer, but also beyond (for detailed overview on post-transcriptional targeting we refer to [51,52]).

#### 2.4.3. Focus on the AAV Capsid—Achieving Specificity by Transductional Targeting

Transductional targeting aims to re-direct particle uptake and thus differs substantially from the other two targeting strategies where particle distribution is not affected, but transgene expression is restricted. In transductional targeting, the vector distribution is modified as the particles are blinded for binding to receptors naturally used for cell attachment and internalization. Furthermore, AAV vector particles are equipped with a targeting moiety such as a receptor binding peptide that replace the function of the natural receptor binding motifs in cell binding and entry. Ideally, unspecific uptake is also avoided by impairing the binding of AAV particles to glycan residues such as the above-mentioned HSPG. As a consequence, less vectors are required, and the risk of side effects induced by high antigenic load, off-target uptake, or off-target expression is substantially lowered. For AAV, non-genetic and genetic transductional targeting strategies are developed for various AAV serotypes. Non-genetic approaches do not change the amino acid composition of the capsid proteins. Instead adaptors are covalently or non-covalently linked, which then either directly mediate the novel cell surface binding or serve as acceptors for the actual targeting ligand. Bi-specific antibodies that connect the capsid and a cellular receptor are an example of a non-genetic targeting approach [53]. Alternative strategies use chemical coupling of biotin to amino groups of capsid surface exposed arginine (R) or lysine (K) residues that are subsequently used to bind avidin-linked receptor binding peptides [54]. Although biotin-(strept)-avidin is one of the strongest non-covalent linkages, it cannot be excluded that linker and capsid become dissociated when applied in vivo. The same holds true for the bi-specific antibody strategy. Thus, covalent bonds are currently preferred. In this regard, Mevel and colleagues recently reported on isothiocyanate coupled ligands again using amino groups on the AAV capsid as a landing platform [55].

In genetic targeting approaches, the moiety that mediates cell transduction is inserted into the capsid proteins at the DNA level. This genetic modification needs to be placed at positions that do not impair capsid assembly while being exposed at the capsid surface. As discussed in detail in [10], these criteria are met at the tips of the protrusions of the AAV capsid, located at the 3-fold symmetry axis, as well as the N′ terminus of VP2, the second largest AAV capsid protein. In case residues responsible for natural receptors have been mapped and capsids are blinded by destroying the respective motifs, specificity—or in other words cell-type selectivity—is solely dependent on the specificity of the targeting ligand. This was demonstrated by Munch and colleagues, who blinded the AAV2 particles by amino acid substitutions (capsid amino acid residues R585A, R588A) and used a designed ankyrin repeat protein (DARPin) that recognizes human Her2/neu for particle navigation [56]. In a xenograft mouse model, precise delivery of vector genomes following intravenous injection of these targeting vectors was observed as the DARPin was unable to bind to murine Her2/neu, but did bind the grafted human cells as exclusive targets [56]. Genetic transductional targeting is also possible with nanobodies [57], while single-chain antibodies or full antibodies rely on non-genetic approaches as the cell nucleus, the site of capsid assembly, does not support correct folding of antibodies. Transductional targeting strategies were extremely successful in development of AAV capsid variants for various tumor cell types as well as cells involved in the tumor microenvironment (Table 2). AAV-L14, the first successful example of genetic transductional targeting, gained the ability to transduce mouse melanoma cells via beta 1 integrin, thereby overcoming a block in transducing, for example, B16F10 mouse melanoma cells [58]. Likewise, the insertion of RGD4c peptide into the AAV2 capsid enabled the transduction of tumor cell lines via ανβ3 or ανβ5 integrin receptors [59,60,61] but were also used to target proliferating endothelial cells [61] which are found at the tip of tumor vessels in vivo. Other peptides have demonstrated promise to mediate transduction of glioblastoma cell lines and chronic myeloid leukemia to name a few more examples (reviewed in [10,62,63]). Besides cell lines, also primary cancer cells are targeted. For transducing chronic lymphocytic leukemia cells (CLL), which are refractory to AAV, AAV-Mec A was developed, which uses the GENQARS peptide to target a so far unknown receptor [64]. Further examples are primary breast cancer cells (AAV-RGDLGLS and AAV-ESGLSQS) [65] and primary melanoma cells (AAV-C4 (PRGTNGP) and AAV-D10 (SRGATTT)) [65].

Besides RGD4c, a portfolio of AAV targeting vectors are nowadays available for improved transduction of endothelial cells (reviewed in [10,62,63] and Table 2), some of which are also functional in the context of quiescent endothelial cells or endothelial progenitor cells which are discussed as tools for realizing “Trojan horse” concepts [81]. Moreover, for DC [83] and T cells [56] targeting AAV vectors are available.

Targeting peptides were identified by phage display in the early days of transductional targeting. Such peptides were selected from large phage libraries for target cell binding in panning approaches. However, only parts of the peptides showed promise as AAV targeting ligands. This limitation was overcome by the development of AAV peptide display libraries, which select peptides for suitable targeting in the context of the capsid, in conditions that mirror the later application, and which enable the selection of AAV capsid variants that overcome pre- as well as post-entry barriers [62]. Briefly, AAV peptide display libraries consist of approximately 10 × 10^6^ AAV capsid variants, which differ in peptides that are displayed on their surface. Since they encode for the *cap* sequence including the peptide insertion, each variant can be identified via sequencing. Libraries are used to infect target cells ex vivo or in vivo, and variants that successfully infect the target cell types are amplified and re-applied. Commonly, 3–5 rounds of selections are performed and monitored by next generation sequencing (NGS) to control selection conditions and identify candidates that accumulate in target cell-types and thus most likely possess promising features for the desired application. These candidates are then produced as viral vectors and comprehensively characterized regarding production yield, efficiency, specificity and immune escape capabilities prior to application. In addition, these novel variants can be used as tools to identify the nature of barrier/s that impair transduction by the parental serotypes thereby adding to the understanding of the host-AAV interaction [81,83]. Finally, strategies are available to map the targeted receptors [85].

Furthermore, combinations of genetic and non-genetic targeting strategies have been developed. In those cases, a sequence is introduced that serve as acceptor for the targeting ligand in keeping with the concept of developing “one-fits-all” targeting vectors. Specifically, an antibody binding motif (Z34C) [86], a biotin acceptor sequence [87] or an aldehyde tag [88] was introduced. Following production, such vectors can be equipped for example with antibodies targeting cell surface receptors on tumor cells [86]. As an advantage, developmental turnaround times are low due to the modular setup of the technique regarding the choice of receptor interaction to be targeted; however, “one-fits-all” targeting vectors face the challenge that a suitable ligand–receptor interaction needs to be identified beforehand. In addition, if applied in vivo also in the case of “one-fits-all” targeting vectors (like in case of the non-genetic transduction targeting), strategies that use a covalent attachment are preferred due to their increased stability.

In summary, a portfolio of AAV serotypes and engineered AAV variants are at hand as well as techniques to tailor the AAV vectors system, making it likely that a suitable AAV vector type is (becoming) available for any application.

## 3. Use of AAV Vectors for Cancer Gene Therapy in Preclinical Models

AAV vector-based preclinical approaches can be assigned to the different hallmarks of cancer, related to the target cell population and transgenes used (Figure 2). The different approaches using AAV vectors in preclinical cancer models have been comprehensively reviewed in 2016 by Santiago-Ortiz et al. [11]. Here, we are going to provide an update on the latest developments in this exciting field.

### 3.1. AAV Vector-Mediated Targeting Tumor Cell Proliferation and Tumor Cell Death

#### 3.1.1. Cytotoxic Killing of Tumor Cells: Suicide Gene Transfer

Suicide gene transfer is a method for cancer treatment that relies on a selective conversion of non-toxic compounds into cytotoxic drugs inside cancer cells. While preclinical data have been promising, clinical significance of the approach is still lacking. The most widely used suicide gene, Herpes Simplex Virus thymidine kinase (HSV/TK), induces cell killing in transduced cells by converting ganciclovir (GCV) into the toxic metabolite GCV-triphosphate and was demonstrated to induce bystander toxicity to neighboring tumor cells. AAV vector-mediated HSV/TK gene transfer was studied in a murine breast cancer xenograft model [89], which demonstrated decreased tumor growth. Synergistic tumor inhibitory effects were seen in head and neck cancer models in combination with irradiation [90] and in a bladder cancer model in combination with the overexpression of endostatin, an endogenous inhibitor of angiogenesis [91]. An early study using AAV vector-mediated delivery of HSV/TK together with Interleukin 2 in both immunocompetent and immunodeficient murine hepatoma models demonstrated synergistic effects compared to HSV/TK gene transfer alone. Interestingly, the combination was even more effective in non-GCV-treated animals compared to GCV-treated immunocompetent (C57L/J mice), indicating that immune modulatory effects play an important role [92]. The same group has successfully used transcriptional targeting by AAV vector-mediated HSV/TK gene transfer under the control of an alpha-fetoprotein enhancer and albumin promoter to specifically target AFP-positive human hepatocellular carcinoma cells in a nude mouse model [93]. More recently, Kahn et al. developed a modified production protocol by providing miR-636 during the packaging process. This miRNA was identified based on its ability to bind to the AAV genome (ITRs) and also dysregulate mitogen-activated protein kinase (MAPK) signaling during vector production. AAV6 vectors packaged in this way demonstrated increased transduction efficiency in vitro and expression of an inducible caspase 9 suicide gene [94]. Treatment with a dimerizer drug resulted in increased cell killing in a murine T-cell lymphoma (EL-4) model following intratumoral application [94].

#### 3.1.2. Inducing Tumor Cell Death: Apoptosis and Beyond

Defective apoptosis plays a central role in the development and progression of cancer. Cancer cells downregulate apoptotic programs and gain proliferation and survival advantages through overexpression of anti-apoptotic and/or downregulation of pro-apoptotic proteins. Currently, therapeutics targeting apoptosis have been introduced in the clinic like venetoclax, a small molecule inhibitor of the anti-apoptotic protein Bcl-2, for the treatment of CLL. Gene therapy represents another valid technical approach to tip the balance towards apoptosis induction in tumor cells. Overexpression of pro-apoptotic factors has been extensively studied in the past. TNF-related apoptosis-inducing ligand (TRAIL), a cytokine that binds to certain death receptors, induces apoptosis primarily in cancer cells and was extensively studied in different cancer models in vitro and in vivo using the AAV vector system [95,96,97,98,99]. Importantly, synergistic effects with chemotherapeutics were seen and targeting towards tumor cells was achieved by using the above-mentioned tumor specific promoter human (h) TERT [100,101]. In addition, this approach was successfully used to treat patient-derived orthotopic xenografts of human glioblastoma in mice by systemic application of an AAV9 vector carrying sTRAIL as a transgene [102]. However, despite encouraging preclinical results, these efforts have not yet translated into clinical application. Another approach relied on the AAV vector-mediated overexpression of mutant forms of Survivin in cancer cells. Survivin is overexpressed in a large number of human cancer cells and tissues, and increased expression is associated with apoptosis inhibition, drug resistance, and poor prognosis. Overexpression of the Survivin-Thy34Ala variant induced apoptosis in gastric cancer cells and decreased cancer growth in vivo. Synergistic effects with chemotherapy (i.e., 5-Fluorouracil, oxaliplatin) were reported [103,104] and, beyond apoptosis, the inhibition of angiogenesis was observed in colon cancer (HCT116, HT29) xenograft models [103].

Downregulation of anti-apoptotic proteins using small interfering RNAs (siRNAs) or short hairpin RNAs (shRNAs), or epigenetic modulation of expression levels of pro-apoptotic factors or target receptors represents a novel exciting field of development to therapeutically target apoptosis. Since low transfection efficiency, enzyme degradation, inappropriate subcellular localization and endosomal trapping of siRNAs in cells represent major challenges, viral vectors-based methods to target pro-apoptotic signals using RNAi or sgRNA are being developed. As already mentioned, overexpression of TRAIL is a widely studied approach to target apoptosis. Interestingly, the targeting of hTERT activity, which is overexpressed in many tumor cells and has a key role in self-renewal, was demonstrated to synergize with apoptosis inducing approaches. Based on the identification of a hTERT C-terminal fragment (hTERTC27) that binds to telomers but lacks enzymatic activity telomer-length maintenance can be disturbed [105]. Moreover, an AAV/hTERTC27 vector, expressing this fragment, was applied to HeLa cells, resulting in telomer dysfunction and cell senescence [106]. Finally, RNA interference-based silencing of hTERT, when combined with TRAIL overexpression using an AAV vector-based approach, was shown to restore apoptosis-induction in human oral squamous cells in vitro and in vivo. Mechanistically, TRAIL expression levels were enhanced when hTERT was silenced [107]. In another study, AAV vector-mediated overexpression of miRNA-7 in conjunction with the expression of soluble TRAIL decreased tumor growth in human glioblastoma mouse xenograft models. Mechanistically, miRNA-7 expression in glioblastoma cells resulted in downregulation of the growth-promoting Epidermal Growth Factor Receptor (EGFR) pathway, upregulation of the death receptor pathways and consequently enhanced TRAIL-mediated apoptotic cell death both in vitro and in vivo leading to tumor growth inhibition [108]. Interestingly, it was demonstrated recently that the miRNA221/222 cluster is upregulated in TRAIL-resistant liver cancer cells. AAV vector-mediated overexpression of TRAIL together with an miRNA221zip knockdown construct [109] induced apoptosis in TRAIL-resistant liver cancer cells in vitro and in murine models in vivo [110].

As a further example for an epigenetic treatment approach, downregulation of miRNA-21 using AAV2 vector-mediated stable expression of hairpin RNAi targeting miRNA-21 attenuated growth in different cancer cell lines and in vivo in murine tumor models (HT29 colon cancer and PC3 prostate cancer) [111]. Moreover, downregulation of miRNA-21 in conjunction with miRNA-7 overexpression induced caspase-mediated apoptosis, and synergistic reduction in cell growth and invasion was demonstrated in glioblastoma models [108].

In an siRNA screening approach to search for potential therapeutic targets of basal-like triple-negative breast cancer cells [112], genes related to proteasome activity like Proteasome Subunit A (PSMA) Type 1 (PSMA1), PSMA2, and PSMB were identified to confer vulnerability. AAV vector-mediated expression of an shRNA to downregulate PSM2 resulted in deceased tumor growth in a mouse xenograft model of basal-like triple-negative breast cancer. Mechanistically, targeting the proteasome induced apoptosis by accumulation of NOXA, a protein involved in regulating cell death decision [113].

Based on findings that miRNA-128 expression is decreased in glioma cells, the effects of AAV vector-driven overexpression of miRNA-128 in human glioblastoma models was investigated and glioma cell proliferation was significantly reduced in vitro and in glioma xenograft models in vivo. In this study, miR-128 expression was associated with decreased expression of the oncogene Bmi-1, by direct regulation of the Bmi-1 mRNA 3-untranslated region, through a single miR-128 binding site [114]. Similarly, AAV vector-mediated inhibition of Bmi-1 driven by the expression of a Bmi-1 shRNA suppressed tumor growth and stem cell-like properties of gastric cancer cells in vitro and in vivo [115].

Finally, an interesting approach that aimed to increase the efficacy of chemotherapy was recently reported by Zhen et al. and is based on the overexpression of Decorin (DCN). DCN is a small leucin-rich proteoglycan in the extracellular matrix. It can influence tumor growth via the regulation of several cytokines. AAV vector-mediated overexpression of DCN in neuroblastoma influenced macrophage function and led to an upregulation of Secreted Protein Acidic and Rich in Cysteine (SPARC), which plays an important role in cell growth and angiogenesis. Moreover, uptake of nab-Paclitaxel, an albumin-bound formulation of the chemotherapeutic drug Paclitaxel, is influenced by the albumin binding properties of SPARC. Consequently, AAV vector-mediated DCN overexpression led to increased uptake of nab-Paclitaxel in neuroblastoma cancer cells and combined application further decreased tumor growth in a xenograft model compared to nab-Paclitaxel alone [116].

Pyroptosis has long been regarded as a caspase-1 mediated monocyte death in response to certain bacterial insults. It was identified as an immunogenic variant of apoptosis due to the fact that it is related to Gasdermin-driven pore formation, which leads to the release of immunogenic cell content as well as the release of proinflammatory cytokines (i.e., Interleukin-1β and Interleukin-18) [117]. Overexpression of caspase-1, following injection of an AAV1 vector encoding caspase-1 under a Schwann-cell specific promoter into schwannoma tumors, led to regression of these tumors with essentially no vector-mediated neuropathology in a murine model [118]. More recently, the same group successfully used the same vector to overexpress the N-terminal active fragment of Gasdermin-D [119] as well as the apoptosis-associated speck-like protein [120] in the respective schwannoma model.

Recently, an AAV vector-based strategy was reported for the treatment of Neurofibromin 1 (NF1) related tumors. Aggressive malignant peripheral nerve sheath tumors (MPNST) lack NF1. NF1 suppresses Ras activity via a GTPase-activating protein-related domain (GRD). By using a number of natural AAV serotypes or AAV with engineered capsids (AAV-DJ) to express GRD-constructs, inhibition of the Ras-pathway was achieved in MPNST cells [121].

The E6 oncoprotein of “high-risk” Human Papilloma Virus (HPV) types is involved in the pathogenesis of HPV-associated cancers such as cervical carcinoma. Mechanistically, E6 is involved in the degradation of the tumor suppressor gene p53, thus contributing to carcinogenesis. An AAV2 vector carrying a single guide RNA (sgRNA) targeting E6 was used in cervical cancer cells engineered to stably express CRISPR/Cas9. AAV-sgRNA-E6-transduced cells showed reduced expression of E6, increased expression of p53, increased apoptosis and their growth was suppressed in vitro. In addition, tumor growth was inhibited following the intratumoral application of the vector in a mouse xenograft model of cervical cancer [122].

Bone morphogenetic protein 4 (BMP4) is a member of the TGF-β superfamily of cytokines that affect bone and cartilage formation. Moreover, BMP4 was shown to induce growth arrest and apoptosis in myeloma cells [123]. Recently, AAV8 vectors were used to systemically overexpress BMP4 from the liver in a mouse model of multiple myeloma, in which humanized bone scaffolds were implanted subcutaneously. As soon as BMP4 overexpression became detectable in the blood, myeloma cells were injected into the scaffolds and myeloma growth was inhibited compared to control animals. However, the overexpression of BMP4 resulted in significantly reduced trabecular bone volume in BMP4 overexpressing mice, indicating that this approach is not feasible for further translation [124].

Nuclear protein 1 (NUPR1) is a transcriptional regulator involved in stress response and was found to be overexpressed in a number of cancer entities as well as to play important roles in tumor progression and metastasis [125]. Li et al. used an shRNA-mediated, AAV vector-based approach to inhibit NUPR1 expression in a tumor xenograft model of lung adenocarcinoma (i.e., A549). Moreover, the combination of AAV vector-mediated NUPR1 shRNA expression and trifluoperazine (TFP) treatment, an anti-psychotic drug that binds to NUPR1 and is able to mimic NUPR1 deficiency in cancer cells, showed synergistic growth inhibitory activity. Mechanistically, premature senescence was induced in vitro and in vivo [126].

As mentioned above, targeting strategies have been developed to allow targeted delivery of therapeutic approaches towards tumor cells, while allowing systemic application of the respective AAV vectors. The transcription factor NF-κB is a transcription factor that is centrally involved in innate and adaptive immune functions and in inflammatory response. Since NF-κB is also over-activated in tumor cells, it represents a promising target for cancer therapy that can be exploited via transcriptional targeting. Recently, an AAV vector was constructed carrying an NF-κB specific promoter by fusing an NF-κB decoy sequence with a minimal promoter. The vector was used to restrict CRISPR/Cas9 expression to cancer cells only and to use an sgRNA that targeted telomeric DNA [127] to induce cancer cell death and to inhibit tumor growth in mice following intravenous vector administration.

### 3.2. AAV Vector-Mediated Targeting of Tumor Angiogenesis

The formation of new vascular structures by angiogenesis (i.e., formation of new vascular structures from preexisting ones) and vasculogenesis (i.e., formation of a novel vascular network by bone marrow derived precursor cells) represent physiological processes during embryonic development, wound healing and other regenerative processes. On the other hand, insufficient or abnormal angiogenesis is involved in a large number of pathological conditions, including cancer. With respect to cancer, in the 1970s the American surgeon Judah Folkman coined the term “Angiogenic Switch” to describe a shift of the equilibrium between soluble pro- and anti-angiogenic factors present in the tumor microenvironment towards pro-angiogenic factors, leading to the induction of tumor angiogenesis, thus propagating tumor growth and metastasis [128]. Hypoxia, driven by a defective vasculature in tumors, drives the expression of pro-angiogenic factors produced from the tumor cells, but also from cells in the tumor stroma. This shifted balance results in the formation of a chaotic vascular structure, promoting hypoxia and hampering access for chemotherapeutics to the tumor site by creating an increased interstitial fluid pressure [129]. Therapeutics based on the principle of tumor angiogenesis have been developed during the last decades and are now well-established in the clinical setting for the treatment of a large number of cancer entities. Angiogenesis inhibitors include the monoclonal antibody (mAB) bevacizumab, which targets the pro-angiogenic factor vascular endothelial growth factor (VEGF), the mAB ramucirumab, which targets VEGF Receptor 2 (VEGFR2) and the fusion construct VEGF-Trap (aflibercept) that consists of domains derived from VEGFR2 and VEGFR1, and thus binds the VEGF isoforms VEGFA and VEGFB as well as Placental Growth Factor. Furthermore, a large number of different tyrosine kinase inhibitors (TKI) have been developed (reviewed in [130]). TKI-like pazopanib, sunitinib and many others, target different angiogenic pathways (i.e., VEGFR, Platelet Derived Growth Factor Receptor (PDGFR) and Fibroblast Growth Factor Receptor (FGFR)) amongst others (reviewed in [131]). Since the half-life of antibodies or fusion constructs in the circulation is limited (i.e., 20 days for Bevacizumab, 30h for Pazopanib), frequent re-applications are required. Gene therapeutic approaches to target tumor angiogenesis were proposed soon after the principle was shown to be efficacious in the clinical setting. Beyond targeting angiogenic pathways by overexpression of anti-angiogenic proteins or antibody constructs or repressing the expression of pro-angiogenic factors, direct targeting of endothelial cells and other related cells is another interesting approach, which can, in principle, be addressed by gene therapy (see Table 2 for respective optimized vectors). Cellular abnormalities in tumor blood vessels provide the pathophysiological background [132] for such approaches. Moreover, with the success of antibodies targeting immune checkpoints in the treatment of cancer, it has become clear that blocking tumor angiogenesis synergizes with immunotherapeutic approaches [133,134], providing room for new developments in this field.

#### 3.2.1. Overexpression of Anti-Angiogenic Factors

Initially, AAV vectors were used to overexpress physiological anti-angiogenic factors such as endostatin [135,136,137,138,139,140] or angiostatin [141,142] or combinations of endostatin and angiostatin [143]. Moreover, combinations of an endostatin variant (endostatin P125A: substitution of proline by alanine at position 12) with chemotherapeutics [144,145] showed synergistic tumor growth inhibition in ovarian and breast cancer models. Moreover, the combination of endostatin overexpression with a suicide gene approach in bladder cancer led to decreased tumor growth in these models [146]. Thrombospondin-1 is an anti-angiogenic protein that possesses three type I repeats (3TSR) near the center of the protein and a CD47-binding peptide (CD47) in its C-terminus. It has been reported that treatment with recombinant 3TSR induces tumor regression, normalization of the tumor vasculature, and improves uptake of chemotherapy drugs in an orthotopic, syngeneic mouse model of advanced stage epithelial ovarian cancer [147]. In addition, AAV vector-mediated expression of 3TSR resulted in abrogation of angiogenesis and inhibition of tumor development in a mouse model of ovarian cancer [148]. Ang-(1-7) represents an endogenous heptapeptide hormone of the renin-angiotensin system, which was shown to exert pleiotropic antitumor effects in lung cancer models mediated by inhibitory effects on DNA synthesis, tumor cell migration and invasion as well as anti-angiogenic activity. Overexpression of Ang-(1-7) using AAV8 vectors via tail vein injection in a lung cancer model reduced angiogenic factors and tumor vascular density and inhibited tumor growth [149]. Similarly, the approach was effective in a nasopharyngeal carcinoma model [150] and an HCC model [151].

Other anti-angiogenic factors that were overexpressed by the AAV vector system to decrease tumor growth in animal models were vastatin, an anti-angiogenic polypeptide, in a rat HCC model [152], human plasminogen kringle domains by AAV8 vectors in murine melanoma models, and pigment epithelium-derived factor (PEDF) in various models [153,154,155,156,157]. Moreover recently, an AAV3 vector carrying HSV/TK together with the angiogenesis inhibitor kallistatin was constructed. In this case, TK was used as a reporter gene to monitor treatment success by Positron Emission Tomography (PET). Following tail vein injection, mRNA and protein expression of HSV-TK and kallistatin were measured in subcutaneous HepG2 hepatoma tumor xenografts, and in conjunction with the PET data indicated HepG2 tumor targeting [21]. Finally, an AAV2 capsid variant with tyrosine to phenylalanine substitutions, improved for microvascular endothelial cells, was used to express siRNAs targeting the unfolded protein response, which resulted in inhibition of angiogenesis and tumor growth in a breast cancer model [158].

#### 3.2.2. Strategies to Block the VEGF Pathway

AAV vector-mediated long-term blockade of the VEGF-pathway represents another well-studied area of research. An AAVrh.10 vector, derived from a rhesus monkey serotype10 AAV, was used to deliver a single-chain antibody (scAB) containing bevacizumab-derived sequences (AAVrh10.BevMab). Long-term high expression levels following intraperitoneal application were achieved and resulted in significantly reduced ovarian cancer growth through the inhibition of angiogenesis both alone and in combination with chemotherapy [159]. Similarly, direct administration of the AAVrh.10BevMab vector to the CNS of immunodeficient mice implanted orthotopically with human Glioblastoma Multiforme (GBM) cell line (U87G) or primary GBM tumor cells, led to expression of a bevacizumab scAB, and resulted in reduced tumor blood vessel density in the area of the tumor and in a significant reduction in tumor growth [160]. In another preclinical study, intravenous injection of AAV2 vectors was used to deliver the fusion construct VEGF-Trap for long-term expression in vitro and in vivo. With this approach, local tumor growth and metastasis of breast cancer was reduced in a murine model [161]. Moreover, the same approach was used in a rat GBM model in combination with temozolomide, a chemotherapeutic drug widely used for GBM treatment. Again, a single tail vein injection of AAV2-VEGF-Trap [162] resulted in long-term expression and synergistic growth inhibitory effects were seen in combination with chemotherapy. Finally, anti-angiogenic effects were monitored by Magnetic Resonance Imaging (MRI) [162]. A similar approach was performed in a triple negative breast cancer model (MDA-MB-231). Intravenous application of AAV2-VEGF-Trap reduced tumor growth and affected angiogenesis, as measured by MRI parameters of diffusion and a combination of paclitaxel and AAV2-VEGF-Trap demonstrated synergistic growth inhibitory effects [163]. Expression of soluble VEGF-receptor 3 (VEGF-R3) is predominantly involved in lymphangiogenesis and represents another targeted anti-cancer approach that can be applied via AAV vectors [164].

### 3.3. Use of AAV Vectors for Immunotherapy of Cancer

#### 3.3.1. AAV Vector-Mediated Modification of Immune Cells: CAR-T Cells and Beyond

Immunotherapy with T cells expressing chimeric antigen receptors (CAR) is a novel strategy for the treatment of relapsed/refractory B-ALL and aggressive B-cell lymphomas [165]. Applications of this technology for other cancer types and solid tumors are under development [166]. Currently applied methods for the modification of T cells, however, pose at least a theoretical risk of insertional oncogenesis because lentiviral and retroviral vectors integrate the CAR transgene in a semi-random fashion [167]. From this background, the use of AAV vectors for the generation of CAR-T cells is very promising.

In a study from Sather et al., primary human hematopoietic cells were gene edited in the CCR5 locus by introducing mutations using megaTAL nucleases and via AAV vector-mediated delivery providing templates for homology-directed recombination (HDR) into the CCR5 locus. Using this method, they transferred and expressed an anti-HIV-CAR as well as an anti-CD19-CAR with 14% and 9% insertion efficiency, respectively, in primary CD3+ T cells while maintaining CAR-mediated killing activity [168].

Similarly, Hale et al. used megaTAL to disrupt the T cell receptor-α constant (TRAC) locus and inserted an AAV vector-delivered anti-CD19 CAR template. CAR transfer to the TRAC locus by HDR resulted in 42% CD3- CAR expressing PBMCs. They also generated a CAR directed against B cell maturation antigen (BCMA) and inserted it into the TRAC locus to yield 31% CD3- BCMA-CAR expressing cells that produced IFN-γ, IL2 and TNF-α after stimulation [169].

In order to generate allogenic T cells expressing anti-CD19 CAR, MacLeod et al. have used an engineered homing endonuclease that mediates insertion of the AAV vector-delivered CAR donor template into the TRAC locus. They demonstrated that 48.6% of engineered T cells with TCR knockout (CD3-) express CD19-CAR and that these modified cells were functionally active in vitro with regard to antigen specific proliferation, cytolytic activity and release of cytokines, and exhibited potent antitumor responses to CD19-expressing tumor cells in vivo [170].

The CAR-T cell approach currently used in the clinic is only available using autologous cells, which create problems in production and limit the access for patients who have their T cells depleted or stressed due to previous therapies. Techniques to knock-in the CAR into target cells and simultaneously knock-out genes that prevent allogeneic therapy, such as the endogenous T cell receptor, are promising strategies to overcome this barrier [167].

Using the CRISPR/Cas9 system, Eyquem et al. integrated an AAV vector-delivered anti-CD19-CAR into the TRAC locus. CRISPR/Cas9 was transferred together with sgRNA by electroporation and the CAR homology template was delivered by AAV6. Efficient targeting of the TRAC locus was achieved with over 40% CAR insertion rate. In a mouse model of ALL, these CAR T cells showed improved T cell potency compared to retrovirally-generated CAR T cells [171].

In another study, Albers et al. used the CRISPR/Cas9 system and AAV6 vector-mediated transfer of a TCR gene to disrupt the endogenous TRAC locus and simultaneously provide an engineered TCR gene to modify T cell specificity. They reached a targeted transgene integration efficiency into the TRAC locus of 27%. When TCR-engineered T cells were adoptively transferred into NGS mice in an AML xenograft ML-2 tumor model, edited T cells executed significant tumor cell lysis leading to reduced tumor volumes [172].

Donor αβ T cells that are normally removed from the graft used for allogeneic bone marrow transplantation were engineered to express a CD19-specific CAR while simultaneously deleting the endogenous T cell receptor. Specifically, genome editing with Cas9 ribonucleoprotein and AAV6 for template delivery was applied to integrate a CD19-specific CAR in-frame into the TRAC locus. Greater than 90% of cells lost TCR expression, while >75% expressed the CAR. These cells efficiently killed target cells in vitro and in a xenograft model in vivo, without off-target effects or induction of graft-versus host disease [173].

More recently, Dai et al. established a new system to generate CAR-T cells with the newly introduced CRISPR/Cpf1 system in combination with AAV vector-mediated gene transfer. To engineer primary T-lymphocytes, they electroporated cells with Cpf1 mRNA and subsequently transduced them with an AAV vector providing crRNAs (CRISPR RNA) that target both TRAC and programmed cell death 1 (PDCD1/PD-1) gene loci. Using AAV6, both genes were successfully edited with mutation efficiencies of 60.39% for TRAC and 80.07% for the PDCD1 locus in bulk unsorted cells, which was more efficient compared to AAV9-crRNA delivery. In addition to crRNA (TRAC and PDCD1) transfer, AAV vectors were simultaneously utilized to provide CD22-CAR as a template for homology-directed-repair knock-in, which resulted in 44.6% of CD22 CAR-expressing T cells after stimulation. In addition, bi-specific CAR-T cells directed against CD22 and CD19 were also generated by combined transduction of AAV vectors encoding the respective crRNAs and either CD22 or CD19-CARs. Using this approach, 21.7% of double-positive CAR-T cells were achieved. Finally, cells generated with this new system had favorable characteristics over CAR T cells generated by CRISPR/Cas9, such as lower levels of exhaustion markers while maintaining efficient functionality with respect to cancer cell killing and cytokine production [174].

Besides T cells, also other immune cells such as NK cells have been modified using the CRISPR/Cas9 method. Pomeroy et al. established an approach to knock out inhibitory signaling molecules such as PDCD1 in NK cells, which led to enhanced cytokine production and tumor cell killing. When combined with AAV6 vector-mediated delivery of a template for homologous recombination they achieved integration into the adeno associated virus integration site 1 (AAVS1) safe harbor locus in about 75.6% of activated primary human NK cells [175].

#### 3.3.2. AAV-Based Cancer Vaccination Approaches

An antigen-specific cancer vaccination approach was published by Chang et al. who exploited mesothelin (MSLN) as a frequent tumor antigen for ovarian cancer as a cell-based vaccine. Mesothelin-based vaccination (Meso-Vax) and AAV vector-delivered interleukin (IL)-12 showed potent antitumor immune responses such as increased mesothelin-specific CD4+ and CD8+ T cell precursors and high anti-mesothelin antibodies, which resulted in blockade of tumor formation in 100% of mice for at least 60 days after challenge [176]. In a subsequent study by the same group [177], the efficacy of the Meso-Vax and AAV-IL-12 combination for tumor treatment was confirmed. Another vaccination strategy was developed by Li et al. who used AAV vectors to transfer α-fetoprotein (AFP) into DC to generate DC-derived exosomes (Dexs). Vaccination with Dexs facilitated naïve T cell proliferation and the generation of antigen-specific T lymphocytes, which exhibited anti-tumor responses against HCC in vitro. When DC precursors were stimulated with Dexs, specific immune responses against HCC could be induced more effectively [178]. In a study by Mirandola et al., DC were transduced with AAV vectors expressing cancer/testis antigen sperm protein 17 (SP17), which is a target for immunotherapy. Simultaneously, DC were treated with p38 MAPK inhibitor. In a mouse model of ovarian cancer, mice vaccinated with the combination of p38 inhibitor and AAV vector-mediated SP17 expression showed superior survival rates with 95% of mice surviving up to 300 days in contrast to survival rates of 98 days for vaccination with AAV-sp17 transduced DCs and 60 days for AAV control-transduced DCs [179]. CEA is a well-described tumor-associated antigen for several human epithelial cancers and is an interesting target for immunotherapy. Since CEA is a self-antigen, tolerance needs to be broken by therapeutic interventions. A study by Hensel et al. showed that a combination of AAV1 vector expressing CEA and GM-CSF application was able to break tolerance to CEA and significantly improved tumor-free survival in a syngeneic MC38 mouse model compared to GM-CSF treatment with a control AAV vector [180]. A capsid engineered AAV6 vector was used by Krotova et al. to efficiently transduce DCs. This AAV vector-based vaccine induced robust antigen-specific immune responses using Ovalbumin (OVA) as a model antigen. Vaccination with AAV-OVA strongly prevented the spread of tumor metastasis to the lung. In a prophylactic tumor model, vector vaccination significantly reduced tumor growth; however, it did not entirely inhibit tumor development, which was most likely due to loss of the OVA-antigen expression in tumor cells. Moreover, the vaccination with capsid-modified AAV6 expressing well-described endogenous, non-mutated, melanoma tumor-associated antigens Tyr and gp100 was able to overcome tolerance to these non-mutated self-proteins [181]. Similarly, the improved transduction efficiency of capsid engineered AAV6 vectors in DCs and potent inhibition of tumor growth and extension of animal survival in a mouse model were reported when AAV6 vectors were used to express prostatic acid phosphatase (PAP), which is often upregulated in prostate cancer [182].

#### 3.3.3. AAV-Vector Mediated Delivery of Cytokines—Modulating the Immunosuppressive Microenvironment

Cytokine supplementation is another promising option for immunotherapy of cancer. A study by Zhu et al. used AAV vectors to deliver IL-27, which can inhibit tumor growth. In tumor-bearing mouse models, intramuscular injection of AAV vectors led to a noticeable reduction in tumor growth in a set of different tumor models (B16F10, MC38, EO771 and J558). Gene transfer of IL-27 enhanced T cell infiltration into the tumor as well as T cell effector functions and led to a significant reduction in FoxP3+ Treg in tumors to 6.63% compared to 39.8% of FoxP3+ Tregs with control vector. Furthermore, AAV vector-mediated IL-27 expression effectively inhibited tumor growth and extended mice survival when combined with GM-CSF vaccination and IL-27 transfer was also able to break anti-PD-1 resistance in tumor models that were not responding to anti-PD-1 monotherapy, but led to highly reduced tumor growth and even complete rejection when combined with anti-PD-1 treatment. Interestingly, no signs for autoimmunity were observed in this study [183]. The same group further developed this strategy by directly applying the IL-27 expressing AAV vector into the tumors. Similarly, they detected strong anti-tumor immune responses, infiltration of effector cells such as CD8+ T cells and a systemic reduction in Tregs. Noteworthy, in mice that had entirely rejected tumors, IL-27 serum levels were significantly reduced even to levels below detection. This strategy limits the systemically available IL-27 and thereby reduces potential risks of toxicity associated with systemic cytokine delivery [184]. IFN-α is a cytokine well known for anti-tumor effects and is already approved for cancer treatment. However, only moderate efficacy and high toxicity were observed with this type of treatment. Vasquez et al. have constructed an AAV8 vector that expresses IFN-α fused to apolipoprotein A-1, which simultaneously facilitates longer half-life of fused IFN-alpha and reduced toxicity. Using an MC38 tumor model, they found 43% of all mice successfully rejected tumors completely without any detectable toxicity. However, the efficacy of this fusion protein-expressing AAV vector was limited by significantly reduced activation levels of CD8+ T cells as measured by granzyme B expression compared to AAV vector expressing only IFN-α. Correspondingly, regulatory T cells were upregulated in spleen and tumors compared to AAV-IFN-α single treatment [185].

In another combination approach, the HCC cell line Hep3B was transduced by an AAV2 vector to express IFN-γ and another AAV2 vector was used to transduce DC for expression of the above shortly mentioned α-fetoprotein (AFP), an HCC related tumor antigen. Expression of AFP in DCs provoked an effective AFP-specific MHC class I restricted CTL response. Together with AFP expression in DCs, IFN-γ expression from IFN-γ encoding AAV vectors, which led to the upregulation of HLA-A2 in Hep3B, further enhanced the sensitivity of AFP-specific CTL responses [186,187]. Intracranial injection of single stranded AAV vectors for the expression of IFN-β for local gene therapy was used both in orthotopic syngeneic (AAV9) and orthotopic xenograft (AAVrh8) glioblastoma mouse models and combinations with the chemotherapeutic drug temozolomide were studied. The survival of mice was prolonged and synergy was observed in combination with temozolomide in the groups receiving temozolomide 3 days after vector administration [24].

Recently, another interesting approach was reported that used an adipose tissue targeting AAV vector to overexpress the proinflammatory cytokine IL-15 and IL-15 receptor α in the abdominal fat pad of mice following intraperitoneal injection. The hypothesis behind this approach was that activation of IL-15 signaling in adipose tissue might lead to NK cell activation and proliferation. Indeed, expansion of NK cells was achieved in normal mice in the adipose tissue and the spleen and tumor growth was suppressed in a subcutaneous murine Lewis lung carcinoma model and a metastatic B16F10 model [188].

#### 3.3.4. Other AAV Vector-Based Approaches for Cancer Immunotherapy

Cancer treatment with immune checkpoint inhibitors has a great potential, however, immune-related adverse events can also occur. To specifically deliver immune checkpoint inhibitors to the tumor site, Reul et al. used the above-mentioned Her2/neu-directed AAV vectors [56] to target Her2/neu expressing cells in Balb/c mice for tumor-specific delivery of immune checkpoint inhibitor anti-PD-1 single chain variable fragments (scFv) after intravenous injection and showed reduced off-target expression in the liver. In this proof-of-concept study, a mild trend for reduced tumor volumes was observed when AAV vectors expressing anti-PD-1 scFv were combined with chemotherapy [189].

In a recent study, Simões et al. elaborated on the function of high circulating serum levels of CD6, which is a co-receptor for the TCR, as a decoy receptor for an immunomodulatory strategy in cancer therapy. CD6 expression in mice was mediated by AAV8 vectors and led to reduced formation of lung metastasis and increased survival after intravenous injection, but failed to significantly impair growth of subcutaneously implanted B16F10 tumor cells [190]. Snyder et al. developed an approach for tumor immunotherapy that includes the induction of necroptosis, which can be a more immunogenic form of programmed cell death than apoptosis. When AAV vectors were used to deliver and express a constitutively active form of receptor-interacting protein kinases RIPK3, which induces necroptotic cell death directly in tumor cells, improved intratumoral concentrations of IFN-γ, significantly reduced tumor growth and extended survival was observed in a B16F10 tumor model. Moreover, in combination with anti-PD1 treatment, AAV vector-mediated intratumoral RIPK3 expression resulted in further improved tumor rejection and overall mice survival [191].

In order to further promote immunotherapy for glioblastoma, Ye et al. developed a new in vivo screening platform based on the CRISPR/Cas9 system using hybrid AAV vectors that encode Sleeping Beauty transposase and guide RNAs. Editing primary T cells allows for the screening of membrane targets that lead to ameliorated anti-tumor function/efficacy. Using this system, they identified *Pdia3* as a candidate gene, the disruption of which boosted CD8+ T cell activity as measured by elevated granzyme a, b and c levels. In addition, in a GBM mouse model, mice receiving T cells with AAV vector-mediated *Pdia3* knockout had significantly reduced tumor volumes compared to a vector control group [192]. In perspective, identification of other targets from similar screens may provide different routes for improving T-cell-based immunotherapy for GBM and potentially more broadly for other cancer types [192].

## 4. Future Perspectives and Conclusions

AAV vector-based cancer treatment strategies have not yet been translated into the clinic. However, very promising approaches have been identified in preclinical models that clearly have the potential to be studied in clinical trials and to be translated in the near future. Improvements in the field of AAV vector technology and techniques for transcriptional, post-transcription and transductional targeting as well as gene editing techniques play central roles in this respect. Of particular importance with regard to safety and efficacy is the potential these novel techniques possess for a precise treatment of the malignancy, lacking off-target effects. It appears that first generation AAV vectors are especially suitable for targeting liver and CNS related cancers for tumor cell-directed treatments due to the favorable gene transfer efficacy of certain natural AAV serotypes. Moreover, systemic applications can be envisioned to treat CNS tumors due to the ability of AAV9 (and distinct further serotypes) to cross the BBB (albeit a combination with transcriptional targeting might be required). Further targets can now be addressed by tailoring natural serotypes for example by capsid engineering approaches as discussed above, resulting in AAV variants that overcome pre- as well as post-entry barriers, thereby enabling the transduction of cancer cells or cells of the tumor microenvironment which are resistant to natural serotypes. In addition, transduction efficiency of cell-types already susceptible, can be further increased. In addition, novel approaches in the field of cancer immunotherapy are currently evolving. Following the development of CAR-T cell therapy and its introduction in the clinic, AAV vector-based techniques to transduce T cells as well as other immune cells like DC or NK cells are emerging and AAV vectors targeting DCs will be of special interest for the development of innovative cancer vaccine approaches including in vivo applications. Finally, intratumoral tumor therapy is very promising, and oncolytic virotherapy has already made its way to the clinic (i.e., HSV-based talimogen laherparepvec (T-VEC) for the treatment of melanoma). While AAV by nature has no oncolytic activity, modifications of the capsid and of the transgene cassette may render this vector more immunogenic, and novel strategies to express factors in tumor cells that induce immunogenic forms of apoptosis are of special interest for the development of AAV vector-based intratumoral tumor therapies in the future.

## Figures and Tables

**Figure 1 cancers-12-01889-f001:**
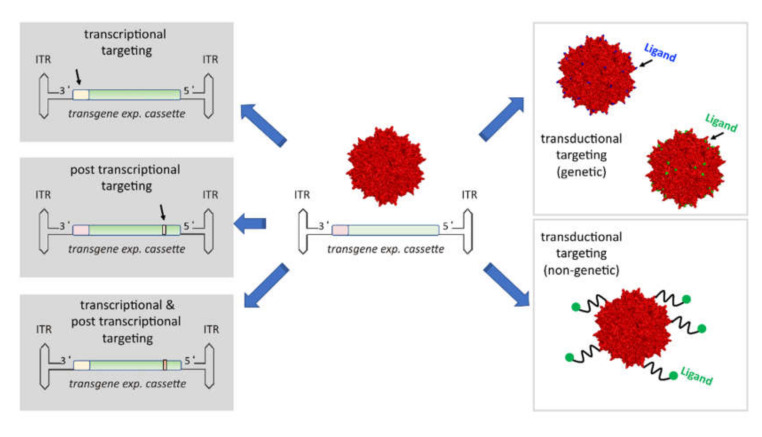
**Targeting strategies.** In order to restrict transgene expression to distinct cell types, i.e., in the context of cancer gene therapy towards the malignant cell, distinct cell types of the tumor stroma or cells from which factors are secreted like muscle cells or hepatocytes so called targeting strategies have been developed. Of note, targeting strategies focusing on the vector genome can be combined with strategies focusing on modifying the capsid. Vectors based on natural serotype capsids do show a broad tropism albeit tissue preferences such as those discussed for the liver are present. By combining destruction or shielding of natural receptor binding epitopes on the capsid with the introduction of—for the AAV—“foreign” molecules that mediate cell binding and internalization, tropism is re-directed towards a novel vector–receptor interaction. This also impacts on biodistribution, is expected to lower the risk of unspecific uptake in off-target tissues and anti-AAV immune responses. This targeting strategy—known as transductional targeting—is either inserting ligands or cell surface receptor binding molecules genetically or via covalent or non-covalent coupling to the AAV capsid. Both transcriptional and post-transcriptional targeting strategies modulate production of the transgene product, i.e., these strategies do not impact on vector biodistribution. Specifically, in transcriptional targeting approaches natural or synthetic promoters are used to restrict transgene expression in a way that ideally no off-target expression takes place. Alternatively, or in combination with transcriptional targeting, post-transcriptional targeting strategies are applied. In post-transcriptional targeting strategies microRNA targeting sites are used, which when recognized by the corresponding microRNA are leading to the degradation of the microRNA target sites containing mRNA. Since microRNAs are frequently downregulated in malignant cells, post-transcriptional targeting strategies are successfully applied to avoid expression in hepatocytes or other healthy cell types that are prone to damage when expressing the therapeutic transgene. ITR = inverted terminal repeats. Vector genomes are shown schematically with ITRs, promoter and transgene plus poly A sequences. Arrow marks position of promoter or microRNA targeting sites, respectively. Promoter: red = ubiquitously active; yellow = restricted activity. Positions marked in blue on the capsid refer to the tip of the highest capsid protrusion, while green shows the tip of the second highest protrusion. Both peaks are used for genetic insertion of peptide ligands (see main text for details).

**Figure 2 cancers-12-01889-f002:**
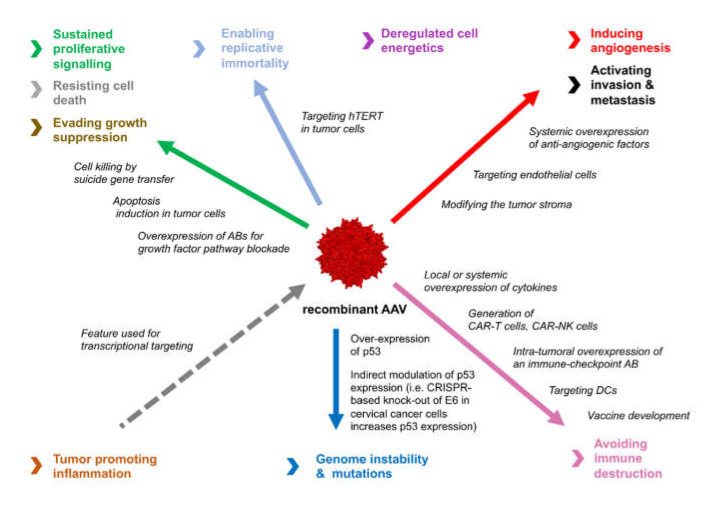
Key strategies of AAV vector-based cancer therapy, applied in preclinical studies so far, are summarized according to the hallmarks of cancer they are addressing. While the interest in anti-angiogenic treatment strategies, which have been a main focus of AAV vector-mediated cancer gene therapy during the last two decades is decreasing, according to the number of newly published papers, the interest and efforts to use AAV vectors for cancer immunotherapy is steadily increasing. Both the development of novel AAV vector-based vaccine strategies and AAV vector-based ex vivo and potentially in vivo modification of immune cells represent promising new avenues for future developments.

**Table 1 cancers-12-01889-t001:** Hallmarks of cancer and key examples of clinically available treatment strategies.

Hallmark of Cancer.	Therapeutic Principle
Evading growth suppression	Cyclin-dependent kinase inhibitors: CDK4/6 inhibitors
Sustained proliferative signaling	Blocking growth factor receptor pathways (i.e., EGFR and others): monoclonal antibodies, tyrosine kinase inhibitors
Resisting cell death	BH3-mimetics that specifically bind to the hydrophobic groove of BCL-2: venetoclax
Enabling replicative immortality	Inhibition of telomerase activity, induction of senescence in tumor cells: a number of commonly used cancer therapeutics are involved in induction of senescence in cancer cells
Deregulating cellular energetics	Inhibition of aerobic glycolysis, inhibition of isocitrate dehydrogenase (IDH) and others: IDH1 and IDH2 inhibitors
Genome instability and mutation	PARP inhibition to facilitate synthetical lethality in homologous recombination deficient tumor cells: PARP inhibitors
Inducing angiogenesis	Inhibition of angiogenic pathways (i.e., VEGF, PlGF): monoclonal antibodies, fusion constructs, tyrosine kinase inhibitors
Activating invasion and metastasis	Targeting the epithelial- mesenchymal transition (EMT) program (e.g., HGF/cMET inhibition), anti-angiogenic treatment strategies
Avoiding immune destruction	Immune activation: immune checkpoint antibodies (i.e., anti-PD-1, anti-PD-L1, anti CLTA-4)
Tumor promoting inflammation	Selective anti-inflammatory drugs: not yet available clinically

**Table 2 cancers-12-01889-t002:** Peptide insertion into the common VP3 region of AAV capsid proteins for tumor transduction.

Name	Target Cell Type	Serotype	Position	Insert	Reference
rRGD453ko	αν-integrin-positive tumor cells	AAV2	**I-453**	CDCRGDCFC	[60]
A584-RGD4C	αν-integrin-positive tumor cells	AAV2	**I-584**	CDCRGDCFC	[59]
AAV-ΔIV-NGR	CD13-positive tumor cells	AAV2	**I-585**	NGRAHA	[66]
AAV-PTP	Plectin-positive tumor cells	AAV2	**I-585**	KTLLPTP	[67]
AAV6-RGD	tumor cells	AAV6	**I-585**	RGD	[68]
AAV-I-587	β1-integrin positive tumor cells	AAV2	**I-587**	QAGTFALRGDNPQG	[58]
AAV-588NGR	CD13-positive tumor cells	AAV2	**I-587**	NGRAHA	[66]
AAV-MO7A	tumor cells	AAV2	**I-587**	RGDAVGV	[64]
AAV-MO7T	tumor cells	AAV2	**I-587**	RGDTPTS	[64]
AAV-MecA	tumor cells	AAV2	**I-587**	GENQARS	[64]
AAV-MecB	tumor cells	AAV2	**I-587**	RSNAVVP	[64]
rRGD587	αν-integrin positive tumor cells	AAV2	**I-587**	CDCRGDCFC	[60]
AAV-C4	tumor cells	AAV2	**I-587**	PRGTNGP	[69]
AAV-D10	tumor cells	AAV2	**I-587**	SRGATTT	[69]
A588-RGD4C	an integrin-positive tumor cell	AAV2	**I-588**	CDCRGDCFC	[59]
A588-RGD4CGLS	αν-integrin positive tumor cells	AAV2	**I-588**	CDCRGDCFC	[59]
AAV-VTAGRAP	tumor cells	AAV2	**I-588**	VTAGRAP	[70]
AAV-APVTRPA	tumor cells	AAV2	**I-588**	APVTRPA	[70]
AAV-DLSNLTR	tumor cells	AAV2	**I-588**	DLSNLTR	[70]
AAV-NQVGSWS	tumor cells	AAV2	**I-588**	NQVGSWS	[71]
AAV-EARVRPP	tumor cells	AAV2	**I-588**	EARVRPP	[72]
AAV-NSVSLYT	tumor cells (CML)	AAV2	**I-588**	NSVSLYT	[72]
AAV-LS1	tumor cells (CML) CD34^+^ cells	AAV2	**I-588**	NDVRSAN	[73]
AAV-LS2	tumor cells (CML) CD34^+^ cells	AAV2	**I-588**	NESRVLS	[73]
AAV-LS3	tumor cells (CML) CD34^+^ cells	AAV2	**I-588**	NRTWEQQ	[73]
AAV-LS4	tumor cells (CML) CD34^+^ cells	AAV2	**I-588**	NSVQSSW	[73]
AAV-RGDLGLS	tumor cells	AAV2	**I-588**	RGDLGLS	[65]
AAV-RGDMSRE	tumor cells	AAV2	**I-588**	RGDMSRE	[65]
AAV-ESGLSQS	tumor cells	AAV2	**I-588**	ESGLSQS	[65]
AAV-EYRDSSG	tumor cells	AAV2	**I-588**	EYRDSSG	[65]
AAV-DLGSARA	tumor cells	AAV2	**I-588**	DLGSARA	[65]
AAV-NDVRSAN	tumor cells	AAV2	**I-588**	NDVRSAN	[74]
AAV-GPQGKNS	tumor cells	AAV2	**I-588**	GPQGKNS	[74]
AAV1-RGD	tumor cells	AAV1	**I-590**	CDCRGDCFC	[61]
AAV8-ESGLSOS	tumor cells	AAV8	**I-590**	ESGLSQS	[65,75]
AAV-MNVRGDL	endothelial cells	AAV2	**I-453**	MNVRGDL	[76]
AAV-ENVRGDL	endothelial cells	AAV2	**I-453**	ENVRGDL	[76]
AAV-SIG	endothelial cells	AAV2	**I-587**	SIGYPLP	[77]
AAV-MTP	endothelial cells	AAV2	**I-587**	MTPFPTSNEANL	[78]
AAV-QPE	endothelial cells	AAV2	**I-587**	QPEHSST	[79]
AAV-VNT	endothelial cells	AAV2	**I-587**	VNTANST	[79]
AAV-CNH	endothelial cells	AAV2	**I-587**	CNHRYMQMC	[80]
AAV-CAP	endothelial cells	AAV2	**I-587**	CAPGPSKSG	[80]
AAV-V_EC_	endothelial cells	AAV2	**I-587**	VSSSTPR	[81]
AAV-NSSRDLG	endothelial cells	AAV2	**I-588**	NSSRDLG	[82]
AAV-NDVRAVS	endothelial cells	AAV2	**I-588**	NDVRAVS	[70,82]
AAV-NDVRSAN	endothelial cells	AAV2	**I-588**	NDVRSAN	[70]
AAV-DIIRA	endothelial cells	AAV2	**I-588**	DIIRA	[76]
AAV-SYENV	endothelial cells	AAV2	**I-588**	SYENVASRRPEG	[76]
AAV-PENSV	endothelial cells	AAV2	**I-588**	PENSVRRYGLEE	[76]
AAV-LSLAS	endothelial cells	AAV2	**I-588**	LSLASNRPTATS	[76]
AAV-NDVWN	endothelial cells	AAV2	**I-588**	NDVWNRDNSSKRGGTTEAS	[76]
AAV-NRTYS	endothelial cells	AAV2	**I-588**	NRTYSSTSNSTSRSEWDNS	[76]
AAV1-RGD	endothelial cells	AAV1	**I-590**	CDCRGDCFC	[61]
AAV-V	dendritic cells	AAV2	**I-587**	VSSTSPR	[83]
AAV-I	dendritic cells	AAV2	**I-587**	ISSSTAR	[83]
AAV1.9-3-SKAGRSP	fibroblasts	AAV1 (aa 445–568 from AAV9)	**I-590**	SKAGRSP	[84]

VP3 common region: The amino acid sequence of the major capsid protein VP3 is also present at the C-terminal part of VP1 and VP2. The VP3 common region of all capsid proteins form the actual capsid. Insertion position: Peptide insertion C′-terminal of VP1 amino acid residue whose number is given. Only sequence of targeting ligand is shown; for details on linker amino acid sequences flanking peptide insert please see reference. ΔIV: peptide insert replaces wild type sequence in variable region IV. wt = wild type.

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
