# Peer review of "Towards Clinical Implementation of Adeno-Associated Virus (AAV) Vectors for Cancer Gene Therapy: Current Status and Future Perspectives"

_cancers, 2020, doi:10.3390/cancers12071889_

Round 1

Reviewer 1 Report

The authors present a comprehensive overview of the use of AAV in cancer therapy. The review covers AAV transduction, the use of AAV as a gene therapy vehicle to date, the engineering of tumour targeting AAV from transgene design and capsid modification, followed by the preclinical use of AAV in the context of treatment of the “hallmarks of cancer”, cancer vaccination, and immunooncology. The topics covered are broad ranging and the content is sufficiently described to provide a high-level overview of AAV as a cancer gene therapy vehicle. Given the length of the manuscript, sections that are irrelevant to the application of AAV in the context of cancer could be removed or significantly condensed (for instance, sections 2.1, 2.2, and much of 2.3) as they are not necessary for the comprehension of the remainder of the cancer-focused content. Nevertheless, this work would likely be of great interest to readers who are new to the field and would like to see the general strategies applied to date.    

Author Response

We are very grateful for the very positive review of our manuscript. As suggested by the reviewer, we carefully revised section 2 and removed unnecessary parts taking also into consideration the comments of reviewer #2 and reviewer #3.

Reviewer 2 Report

In this review, authors described in detail recent advances in AAV-delivery technology and potential ways of targeting tumor cells as well as cellular and molecular factors of an immunosuppressive tumor microenvironment. Overall, the review is very interesting but there are a few concerns regarding its conceptualization, readability and structuring that should be addressed.

Major points:

  1. The title of the manuscript does not properly reflect the fact that currently no AAV vectors are existing for cancer gene therapy. While clinical trials mostly address neurologic and/or ophthalmologic problems or monogenic lesions in other organ systems, AAV-tools are only applied in pre-clinical cancer models, far apart from clinical translation – as authors themselves state. Therefore, I suggest to modify the title of the manuscript: e.g. “Towards clinical implementation of adeno-associated virus (AAV) vectors for cancer gene therapy: current status and future perspectives”. Of course, other alternative titles can also be used that better describe efforts towards translation of the technology into human clinical settings.
  2.  The review should be looked at to streamline it and make it an easier read (some but not all examples are outlined below). The entire Section 2 is difficult to read. It is composed of a large text mass with only very little coherence and inappropriate (sub)structuring. Please, try to introduce more subheadings into 2.3 by describing more explicitly known tropisms of distinct AAV serotypes; describing work in conjunction with liver, with CNS and other organs to make the text more “digestible”. Simplify sentences and clarify what you are trying to say here. Furthermore, for section 2.5 I would suggest to use a heading “Focus on the AAV vector genome – Achieving specificity by transcriptional and pots-transcriptional engineering”. For section 2.6 I suggest to use a heading “Focus on the AAV capsid – transductional targeting”. Moreover, I suggest to move the sentence from row 245-246 “Besides transcriptional and post-transcriptional…..” to the end of section 2.4. in order to provide the link to the subsequent section 2.5. and 2.6.
  3. There should be some more basic facts explicitly and more didactically mentioned concerning AAV-derived vectors in section 2.1. addressing a) the limited cloning capacity of the vector in comparison to other viral vector tools, which might limit its application; b) application of AAV vector in dividing cells might lead to the dilution/loss of episomal concatemers of AAV DNA in the host cell nucleus through cell divisions, which is relevant in proliferating tumor cells; c) wild-type AAV has the ability to stably integrate into the human chromosome 19. By contrast, in AAV gene therapy vectors, this integrative capacity has been eliminated. So, some functions must be provided “in trans” in order to achieve a replicative lytic cycle, which is relevant when viral stocks are produced, etc.
  4. Subsection 3.1.2 deals with cell death. However, the description starting at row 426 deals with targeting of self-renewal. This is somewhat inconsistent at this place. I suggest to indicate this fact explicitly either with an appropriate subheading or to create a separate section for that.
  5. Expression of soluble VEGF-receptor 3 mentioned as a potential anti-cancer approach via AAV (row 580-582). Can be any reference cited for that?
  6. In subsection 3.3.3 I suggest to supplement the subheading “AAV-vector mediated delivery of cytokines – modulating the immunosuppressive microenvironment” in order to indicate the goal of this approach.

Minor points:

  1. While the authors elegantly provided a description on molecular and cellular aspects of tumor angiogenesis referring back to Judah Folkmann´s conceptual contribution in the 1970’s, some key historical papers on gene therapy were omitted, e.g. Theodore Friedmann’s foundational paper in the field entitled “Gene Therapy for human genetic disease?” in Science (1972). However, this has no negative impact on the quality of this review as the cited references fit well in the context they aimed at describing. 

Author Response

Reviewer #2

In this review, authors described in detail recent advances in AAV-delivery technology and potential ways of targeting tumor cells as well as cellular and molecular factors of an immunosuppressive tumor microenvironment. Overall, the review is very interesting but there are a few concerns regarding its conceptualization, readability and structuring that should be addressed.

Response to reviewer #2:

We are grateful for the very helpful comments of reviewer #2.

Major points:

  1. The title of the manuscript does not properly reflect the fact that currently no AAV vectors are existing for cancer gene therapy. While clinical trials mostly address neurologic and/or ophthalmologic problems or monogenic lesions in other organ systems, AAV-tools are only applied in pre-clinical cancer models, far apart from clinical translation – as authors themselves state. Therefore, I suggest to modify the title of the manuscript: e.g. “Towards clinical implementation of adeno-associated virus (AAV) vectors for cancer gene therapy: current status and future perspectives”. Of course, other alternative titles can also be used that better describe efforts towards translation of the technology into human clinical settings.

Response to reviewer #2:

We are grateful to the reviewer’s suggestion and changed the title accordingly.

  1.  The review should be looked at to streamline it and make it an easier read (some but not all examples are outlined below). The entire Section 2 is difficult to read. It is composed of a large text mass with only very little coherence and inappropriate (sub)structuring. Please, try to introduce more subheadings into 2.3 by describing more explicitly known tropisms of distinct AAV serotypes; describing work in conjunction with liver, with CNS and other organs to make the text more “digestible”. Simplify sentences and clarify what you are trying to say here. Furthermore, for section 2.5 I would suggest to use a heading “Focus on the AAV vector genome – Achieving specificity by transcriptional and pots-transcriptional engineering”. For section 2.6 I suggest to use a heading “Focus on the AAV capsid – transductional targeting”. Moreover, I suggest to move the sentence from row 245-246 “Besides transcriptional and post-transcriptional…..” to the end of section 2.4. in order to provide the link to the subsequent section 2.5. and 2.6.

Response to reviewer #2:

We introduced new subheadings, changed the initial subheadings, and introduced the three targeting strategies at the end of 2.4. In addition, we revised chapters 2 and 3 to simplify the reading.

  1. There should be some more basic facts explicitly and more didactically mentioned concerning AAV-derived vectors in section 2.1. addressing a) the limited cloning capacity of the vector in comparison to other viral vector tools, which might limit its application; b) application of AAV vector in dividing cells might lead to the dilution/loss of episomal concatemers of AAV DNA in the host cell nucleus through cell divisions, which is relevant in proliferating tumor cells; c) wild-type AAV has the ability to stably integrate into the human chromosome 19. By contrast, in AAV gene therapy vectors, this integrative capacity has been eliminated. So, some functions must be provided “in trans” in order to achieve a replicative lytic cycle, which is relevant when viral stocks are produced, etc.

Response to reviewer #2:

We are grateful to the reviewer’s suggestions and included respective topics.

  1. Subsection 3.1.2 deals with cell death. However, the description starting at row 426 deals with targeting of self-renewal. This is somewhat inconsistent at this place. I suggest to indicate this fact explicitly either with an appropriate subheading or to create a separate section for that.

Response to reviewer #2:

We thank the reviewer for this important advice. We have modified the section covering targeting of hTERT in order to point out clearly the mechanism and the fact that we want to stress the synergistic effects in conjunction with pro-apoptotic approaches. The section now reads as follows:

„Interestingly, targeting of hTERT activity, which is overexpressed in many tumor cells and has a key role in self-renewal was demonstrated to synergize with apoptosis inducing approaches.  Based on the identification of a hTERT C-terminal fragment (hTERTC27) that binds to telomers but lacks enzymatic activity telomer-length maintenance can be disturbed {Huang, 2002 #764}. Moreover, an AAV/hTERTC27 vector, expressing this fragment, was applied to HeLa cells, resulting in telomer dysfunction and cell senescence {Gao, 2008 #395}. Finally, RNA interference-based silencing of hTERT when combined with TRAIL overexpression using an AAV vector-based approach was shown to restore apoptosis-induction in human oral squamous cells in vitro and in vivo.“

  1. Expression of soluble VEGF-receptor 3 mentioned as a potential anti-cancer approach via AAV (row 580-582). Can be any reference cited for that?

Response to reviewer #2:

We thank the review for this important hint and have added the respective citation: Lin J, Lalani AS, Harding TC et al. Inhibition of lymphogenous metastasis using adeno-associated virus-mediated gene transfer of a soluble VEGFR-3 decoy receptor. Cancer Res 2005; 65: 6901-6909.

  1. In subsection 3.3.3 I suggest to supplement the subheading “AAV-vector mediated delivery of cytokines – modulating the immunosuppressive microenvironment” in order to indicate the goal of this approach.

Response to reviewer #2:

We thank the reviewer for this suggestion, which clarifies the goal of the approach. Subheading 3.3.3 has been modified accordingly.

Minor points:

  1. While the authors elegantly provided a description on molecular and cellular aspects of tumor angiogenesis referring back to Judah Folkmann´s conceptual contribution in the 1970’s, some key historical papers on gene therapy were omitted, e.g. Theodore Friedmann’s foundational paper in the field entitled “Gene Therapy for human genetic disease?” in Science (1972). However, this has no negative impact on the quality of this review as the cited references fit well in the context they aimed at describing. 

Response to reviewer #2:

We agree with the reviewer that the Science paper is a seminal contribution. However, due to the length of the reference list, we hope it is fine if we are not included further papers.

Reviewer 3 Report

The review by Hacker et al. comprehensively covers the current state of the art of AAV based tumor therapy. It is well written and suitable for readers with an interest in cancer and the application of AAVs. Since the manuscript covers also early contributions to the filed, there are some overlaps with previous reviews. Yet from the perspective of a ‘cancers’ reader, the current from provides a complete picture and is well suited.

Minor remarks:

Paragraph 2.4 is relatively short compared to the rest of the text and seems to introduce paragraphs 2.5 and 2.6. Therefore, 2.5 could be numbered 2.4.1 (in analogy to chapter 3) or paragraph 2.4 could be used as an introduction within 2.5.

Figure 1 as such helps to illustrate the text, but could be improved by incorporating more details mentioned in the text or the figure legend. The figure legend could be more concise.

In Figure 2 the cancer hallmark “genome instability & mutations” is not touched by AAV. The text mentions e.g. AAV-sgRNA-E6, which could be seen as a way to address genome instability albeit indirectly.

Based on their long standing in the field, the authors include several self-references. Next to the et al. cases, the senior author appears in reference 10, 60, 61, 62, 67, 176, 185, 186, 187 as “Buning” and only in reference 112 as “Büning”.

Author Response

Reviewer #3

The review by Hacker et al. comprehensively covers the current state of the art of AAV based tumor therapy. It is well written and suitable for readers with an interest in cancer and the application of AAVs. Since the manuscript covers also early contributions to the filed, there are some overlaps with previous reviews. Yet from the perspective of a ‘cancers’ reader, the current from provides a complete picture and is well suited.

Response to reviewer #3:

We are grateful for the very positive review.

Minor remarks:

Paragraph 2.4 is relatively short compared to the rest of the text and seems to introduce paragraphs 2.5 and 2.6. Therefore, 2.5 could be numbered 2.4.1 (in analogy to chapter 3) or paragraph 2.4 could be used as an introduction within 2.5.

Response to reviewer #3:

We revised the numbering of section 2.

Figure 1 as such helps to illustrate the text, but could be improved by incorporating more details mentioned in the text or the figure legend. The figure legend could be more concise.

Response to reviewer #3:

We revised the figure legend.

In Figure 2 the cancer hallmark “genome instability & mutations” is not touched by AAV. The text mentions e.g. AAV-sgRNA-E6, which could be seen as a way to address genome instability albeit indirectly.

Response to reviewer #3:

Figure 2 has been modified to include the AAV-sgRNA-E6 approach as well as older approaches overexpressing p53.

Based on their long standing in the field, the authors include several self-references. Next to the et al. cases, the senior author appears in reference 10, 60, 61, 62, 67, 176, 185, 186, 187 as “Buning” and only in reference 112 as “Büning”.

Response to reviewer #3:

We are grateful for this hint. We corrected the spelling.

Round 2

Reviewer 2 Report

The authors have sufficiently revised the manuscript.

I have no further suggestions.